# A recombinant gp145 Env glycoprotein from HIV-1 expressed in two different cell lines: Effects on glycosylation and antigenicity

José A. González-Feliciano[1], Pearl Akamine[1], Coral M. Capó-Vélez[1], Manuel Delgado-Vélez[1], Vincent Dussupt[2,3], Shelly J. Krebs[2,3], Valerie Wojna[4], Victoria R. Polonis[2], Abel Baerga-Ortiz[1,5]*, José A. Lasalde-Dominicci[1,6]*

**1** Molecular Sciences Research Center Inc., University of Puerto Rico, San Juan, Puerto Rico, **2** U.S. Military HIV Research Program, Walter Reed Army Institute of Research, Silver Spring, Maryland, United States of America, **3** Henry M. Jackson Foundation for the Advancement of Military Medicine, Bethesda, Maryland, United States of America, **4** Division of Neurology, Internal Medicine Department and NeuroHIV Research Program, University of Puerto Rico, Medical Sciences Campus, San Juan, Puerto Rico, **5** Department of Biochemistry, University of Puerto Rico, Medical Sciences Campus, San Juan, Puerto Rico, **6** Department of Biology, University of Puerto Rico, Rio Piedras Campus, San Juan, Puerto Rico

* abel.baerga@upr.edu (ABO); jose.lasalde@upr.edu (JALD)

**Data Availability Statement:** All relevant data are within the paper and its Supporting Information files.

## Abstract

The envelope glycoprotein (Env) of the human immunodeficiency virus (HIV), has been the primary target for the development of a protective vaccine against infection. The extensive *N*-linked glycosylation on Env is an important consideration as it may affect efficacy, stability, and expression yields. The expression host has been shown to influence the extent and type of glycosylation that decorates the protein target. Here, we report the glycosylation profile of the candidate subtype C immunogen CO6980v0c22 gp145, which is currently in Phase I clinical trials, produced in two different host cells: CHO-K1 and Expi293F. The amino acid sequence for both glycoproteins was confirmed to be identical by peptide mass fingerprinting. However, the isoelectric point of the proteins differed; 4.5–5.5 and 6.0–7.0 for gp145 produced in CHO-K1 and Expi293F, respectively. These differences in pI were eliminated by enzymatic treatment with sialidase, indicating a large difference in the incorporation of sialic acid between hosts. This dramatic difference in the number of sialylated glycans between hosts was confirmed by analysis of PNGase F-released glycans using MALDI-ToF MS. These differences in glycosylation, however, did not greatly translate into differences in antibody recognition. Biosensor assays showed that gp145 produced in CHO-K1 had similar affinity toward the broadly neutralizing antibodies, 2G12 and PG16, as the gp145 produced in Expi293F. Additionally, both immunogens showed the same reactivity against plasma of HIV-infected patients. Taken together, these results support the notion that there are sizeable differences in the glycosylation of Env depending on the expression host. How these differences translate to vaccine efficacy remains unknown.

**Funding:** This work was supported by the National Institutes of Health (NIH) grant R01AI122935 to JAL-D and AB-O. This work was also supported by a cooperative agreement (W81XWH-07-2-0067) between the Henry M. Jackson Foundation for the Advancement of Military Medicine, Inc., and the U. S. Department of Defense (DoD). JAL-D wants to acknowledge the US Civilian Research and Development Foundation (CRDF) for providing the necessary funds (funding agreement # OISE-9531011-NIH) to establish the Clinical Bioreagent Center (CBC). Additional support from NIH grants U54NS43011, S11NS046278, and U54MD007587 to VW enabled the patient sample repository. The authors also acknowledge the Puerto Rico Science, Technology & Research Trust, for financing the acquisition of key instruments.

**Competing interests:** The authors have declared that no competing interests exist.

## Introduction

With 35 million infected individuals worldwide, the human immunodeficiency virus (HIV) remains a major global health concern. Most of the current efforts toward a protective vaccine against HIV center on the envelope glycoprotein (Env), the only protein displayed on the virus surface [1–5]. Env is a two-protein system composed of a monomeric gp120 that binds non-covalently to the trimeric, membrane-spanning gp41 [6,7]. Env-based vaccines are notoriously difficult to produce because of their hydrophobic membrane-proximal regions and their extensive glycosylation [8]. Despite these production hurdles, monomeric gp120 has been produced in large amounts and has been thoroughly tested in numerous vaccine trials, including the landmark RV144 vaccine trial in Thailand [9–11].

After the discovery of patient-derived broadly neutralizing antibodies (bNAbs), which provide cross-clade protection through specific recognition of the viral Env, it was clear that the monomeric gp120 did not contain all the relevant epitopes to elicit broad neutralization, and thus, longer Env constructs that better resemble viral spikes would be needed [12–14]. To satisfy the growing need for longer and more structurally relevant Env constructs, two new families of Env designs have been developed: 1) *uncleaved trimers* arising from the elimination of the protease cleavage site that divides gp120 and gp41. These 'uncleaved trimers' consist of a single chain Env with a sequence that stops short of the membrane-proximal region. Env glycoproteins developed in this category include the uncleaved C97ZA012-gp140 [5], CN54gp140 [15], and CO6980v0c22 gp145 [16]. This family of immunogens has been produced in large scale and tested in rodents and non-human primates. Additionally, some of these constructs are currently in Phase I clinical trials. 2) The *native-like trimers* consisting of gp120 and gp41 genetically fused either by an engineered disulfide bond or by a flexible peptide linker. HIV-1 Env glycoproteins developed in this category include the SOSIP trimers [4,17,18], NFL trimers [19], and the UFO constructs [20]. Native trimers, particularly BG505 SOSIP, have been characterized structurally and conformationally, and are also currently being tested for safety and preliminary efficacy in patients [21–25].

The extensive glycosylation on these trimeric versions of Env (both uncleaved and native-like) remains a major limitation toward their high-yield production. Env contains approximately 27 sites for *N*-linked glycosylation, each of which could be occupied by either an oligo-mannose or a complex glycan, imposing tremendous strain on the protein trafficking machinery of the expression host [3]. It has been generally understood that for an Env vaccine to be effective, its glycan shield should resemble that of the infectious virus [26]. Thus, a number of analytical strategies have been developed to measure the precise chemical nature and distribution of the glycans in Env from viral and recombinant sources [27,28].

The glycosylation of recombinant Env vaccines can be affected by the choice of expression system. This is exemplified in Raska *et al.* where it is shown that the types of glycans displayed on recombinant gp120 vary depending on the expression system [28]. This observation was confirmed by Yu et al., who reported that the gp120 produced in CHO-K1 cells is substantially more acidic than the same protein produced in 293F cells due to a higher level of decoration with sialic acid, although the effect of this difference on antibody binding was not entirely clear [29]. Similar glycosylation studies have been performed for native-like trimers, particularly for the SOSIP versions produced in CHO-K1 or in HEK293 cells, revealing minimal differences in the complex vs. mannose glycan distribution between the two hosts [30]. From these detailed glycosylation studies it was concluded that the gp120 portion of the SOSIP trimer contains more high-mannose glycans than the gp41 portion, and that those trimers made in CHO-K1 cells contain a higher level of complex glycosylation than the cleaved trimers

produced in 293 cells [31]. Such detailed glycosylation studies have not been carried out for any of the other native-like trimers (NFL or UFO), or for the uncleaved Env constructs.

In this work, we compared the overall glycosylation and antigenicity of the *uncleaved* CO6980v0c22, a subtype C gp145, produced in CHO-K1 and Expi293F (HEK 293-derived cells). This specific Env construct is currently undergoing clinical testing for safety and immunogenicity in uninfected healthy adults in the United States (ClinicalTrials.gov). Our results show considerable differences in the gp145 glycosylation pattern depending on the cell host. These differences in glycosylation, however, do not seem to greatly affect the binding affinity of bNAbs or reactivity against antibodies from HIV-infected patients.

## Materials and methods

### Antibodies and HIV-1 immunogens

All bNAbs were obtained from the NIH AIDS Reagent Program, Division of AIDS, National Institute of Allergy and Infectious Diseases, National Institutes of Health. The HIV-1 CO6980v0c22 was produced by transient transfection of Expi293F cells and purified by a *Galanthus nivalis* lectin (GNL) affinity column followed by Q-sepharose chromatography. The CHO-K1-produced gp145 was purified following an identical protocol as previously described [16]. Briefly, the culture supernatant was clarified by centrifugation and concentrated by tangential-flow filtration followed by GNL affinity and Q-Sepharose fast flow. The protein was then further concentrated, buffer exchanged into phosphate-buffered saline (PBS), and sterile filtered. Aliquots obtained at 1 mg/mL in phosphate-buffered saline from Advanced Bioscience Laboratories (ABL Inc.) and the U.S. Military HIV Research Program (MHRP), respectively.

### Glycan analysis by MALDI-ToF mass spectrometry

Enzymatic release of *N*-linked glycans from HIV-1 gp145 was carried out following the method of Küster *et al.* [32] with minor modifications. Briefly, excised bands containing 20 μg of gp145 were incubated with 5000 units of PNGase F (NEB) for 24 hrs. at 37˚C [32]. For the analysis of oligosaccharides containing sialic acid, 20 μg of gp145 in solution were incubated with 50 units of α2–3,6,8,9 NEUA (NEB) for 24 hrs. at 37˚C, prior to PNGase F digestion. Subsequently, glycans were extracted into LC-MS water in an ultrasonic bath. The extracted glycans were then dried in a vacuum concentrator and dissolved with water. For MALDI-ToF analysis, glycans were co-crystallized with a solution of 50% acetonitrile/50% 20 mM ammonium citrate containing 134 mM of 2′,4′,6′-Trihydroxyacetophenone monohydrate (Sigma-Aldrich). MALDI-ToF MS analysis of the *N*-glycans was performed in the reflector positive and negative ion mode using a 4800 Plus MALDI ToF/ToF Analyzer (AB Sciex) that was calibrated externally using the Calmix 5 Opti-ToF High-Resolution TIS Calibration Insert (AB Sciex). *N*-glycan assignments were carried out using GlycoWorkbench software and confirmed by MALDI-ToF post-source decay spectra (S2 Fig) [33].

### Peptide mass fingerprinting

The in-gel tryptic digestion of the HIV-1 gp145 protein was performed following the method of Shevchenko et al. with minor modifications [34]. Briefly, 20 μg of gp145 were separated in a 10% Mini-Protean TGX Gel (BioRad). The SDS-PAGE gel was then stained for 1 hr. with Coomassie Brilliant Blue R-250 dye and destained overnight in 50% methanol and 10% acetic acid. Subsequently, the gel band corresponding to the molecular weight of gp145 was excised and destained in 25 mM ammonium bicarbonate diluted in 50% acetonitrile at 37˚C. Then, samples were digested overnight at 37˚C with PNGase F (NEB). After the *N*-linked glycans were

removed, gel pieces were reduced for 40 min at 37˚C with 10 mM dithiothreitol (Millipore-Sigma) diluted in 50 mM ammonium bicarbonate (Fluka). Next, alkylation was performed in the dark with 55 mM iodoacetamide diluted in 50 mM ammonium bicarbonate at room temperature (RT). The samples were then digested with 100 µL of proteomic grade trypsin (17 ng/µL) in 10 mM ammonium bicarbonate containing 10% (vol/vol) acetonitrile at 37˚C overnight. The peptides were extracted with 200 µL of extraction buffer [1:2 (vol/vol) 5% formic acid/acetonitrile] for 15 min at 37˚C. The obtained peptides were dried in a speed vacuum concentrator and resuspended in 40 µL 5% formic acid (Honeywell). To analyze the samples by MS, a mixture of peptides and the MALDI matrix solution [1:1 (vol/vol) sample/72 mM 4-Chloro—α-cyanocinnnamic acid, 70% acetonitrile, and 1.5% formic acid] was spotted onto the MALDI plate insert. MALDI-ToF MS data were acquired in the reflector positive ion mode using a 4800 Plus MALDI ToF/ToF Analyzer (AB Sciex) that was externally calibrated using the Calmix 5 Opti-ToF High-Resolution TIS Calibration Insert (AB Sciex). The mMass—Open Source Mass Spectrometry Tool (Version 5.5.0) was used for spectra and sequence coverage analysis.

## Capillary isoelectric focusing

The gp145 produced in CHO-K1 and Expi293F cells was prepared by mixing 25 µg in 100 µL IEF solution containing 0.8% pH 2–9 Servalyte, 4.22/9.46 pI markers, 4.5 M urea, and 0.35% methylcellulose (MC). The samples were vortexed for 10 s and centrifuged at 14,000 rpm for 10 min at RT. Subsequently, 80 µL of the spun sample was transferred to a 2 mL vial containing a 300 µL vial insert. IEF separation of gp145 immunogens was carried out using an imaging cIEF technique on an iCE3™ Analyzer (ProteinSimple). The PrinCE autosampler temperature was set at 10˚C and the separation was performed in a 50 mm, 100 µm (I.D.) fluorocarbon-coated capillary cartridge (cIEF cartridge fluorocarbon-coated). The solutions containing 0.1 M NaOH in 0.1% MC and 0.08 M phosphoric acid in 0.1% MC were used as the catholyte and anolyte, respectively. The instrumental settings for the gp145 immunogens were as follows: prefocus 1 min at 1.5 kV and focus at 3 kV for 7.0 min. The absorbance at 280 nm was monitored using a whole column ultraviolet absorption detector. The resulting electropherograms were calibrated using the iCE3 CFR control software V4.1 (ProteinSimple) and plotted using GraphPad Prism (Version 6.07).

## Binding kinetics of HIV-1 gp145 to 2G12 and PG16 bNAbs

The binding kinetics of gp145 immunogens to 2G12 and PG16 bNAbs were measured using the Octet QK$^e$ BLI system (Pall ForteBio Corp). The 2G12 and PG16 bNAbs in 0.5X Kinetics Buffer (PBS with 0.02% Tween and 0.05 mg/mL BSA) were loaded onto an anti-Human IgG Fc Capture sensor (Pall ForteBio Corp). Binding kinetics were determined using serial dilutions of CHO-K1-produced and Expi293F-produced gp145. The BLI method for 2G12 was as follows: 2G12 (0.2 µg/mL) load: 500 s at 1200 rpm; baseline: 200 s at 1200 rpm; association: gp145 serial dilutions 100–3.125 nM in 0.5X Kinetics Buffer 900 s at 1200 rpm and dissociation: 1000 s at 1200 rpm. The BLI method for PG16 was as follows: PG16 (0.357 µg/mL) load: 600 s at 1200 rpm; baseline: 200 s at 1200 rpm; association: gp145 serial dilutions 125–31.25 nM in 0.5 X Kinetics Buffer 900 s at 1200 rpm and dissociation: 1000 s at 1200 rpm. To correct for the instrument drift, a reference well containing only 0.5X Kinetics Buffer was monitored in parallel and used for background subtraction. Curve fitting analysis was performed using the global fitting and 1:1 binding model. The mean of $k_a$ (rate of association), $k_d$ (rate of dissociation), and $K_D$ (dissociation constant) were calculated using the ForteBio Analysis 8.2 program.

Binding kinetics for CD4, b6, VRC01 [35], 2G12, PG9, PG16, 2158, 447-52D, PGT121, and 4E10 antibodies were measured using the Octet Red 96 system with the following assay orientation: Ligand: biotinylated gp145; Analyte: HIV-1 antibody. The biotinylated CO6980v0c22 Env was immobilized on streptavidin biosensors at a single concentration of 10 μg/mL for 250 s, to reach ~50% saturation of the biosensor. After reaching baseline, sensors were dipped into six, two-fold dilutions of the monoclonal antibodies at various concentrations for association. Sensors with analyte-ligand complexes were then moved into kinetics buffer to measure dissociation rates. Association and dissociation rates were measured in real-time and were calculated using the ForteBio Analysis software, which fit the observed global binding curves to a 1:1 binding model. Non-specific binding of the monoclonal antibody to the sensor and/or a buffer only reference was subtracted from all curves. Rate constants were calculated using at least three different concentrations of analyte, to achieve an $X^2 < 3$ and $R^2 > 0.90$.

### Enzyme-linked immunosorbent assay (ELISA)

To determine whether differences in glycosylation affect antibody recognition, we investigated the immunoreactivity of IgG from plasma of HIV-positive patients toward CHO-K1-produced and Expi293F-produced gp145. Twenty samples (15 HIV-seropositive and 5 control women; 1/10,000 dilution) were obtained from the repository of the Hispanic/Latino Longitudinal HIV-seropositive women and incubated for two hours on Nunc-Immuno MicroWell plates (Sigma-Aldrich) coated with gp145 produced in either CHO-K1 or Expi293F cells (final concentration 0.5 μg/mL). IgG antibodies were detected with an anti-human HRP-conjugated antibody (1/5000 dilution) and absorbance read at 490 nm. Each sample was run in triplicate and values were normalized against the blank. A Two-Way ANOVA with a Dunnett's multiple comparison test was used to assess statistical differences. Values with a $p < 0.05$ were considered significant.

### Participants' description

Blood samples of 20 participants, 15 HIV-seropositive and 5 control women, were obtained from the repository of the Hispanic/Latino Longitudinal HIV-seropositive women cohort (20 plasma samples) (IRB protocol 1330107). The inclusion criteria included consenting adults with or without HIV infection and without active systemic infections. All participants consented to have samples stored in the cohort repository for future related studies toward the understanding of HIV infection mechanisms and future treatment modalities. Characteristics of the participants are described in Table 1. The HIV-seropositive group was further divided into those who received no antiretroviral treatment (ART, n = 4), used older ARTs (from 2006–2012, n = 7), and those who used newer ART ($\geq$2012, n = 4).

## Results

### Confirmation of HIV-1 gp145 protein identity

The identity of CO6980v0c22 gp145 produced in CHO-K1 and Expi293F cells was confirmed by peptide mass fingerprinting (Fig 1A and 1B). Briefly, the excised protein gel band corresponding to gp145 (S1 Fig) was first PNGase F-digested and then trypsin-digested. The deglycosylated peptides were analyzed by MALDI-ToF. MS results showed an identical distribution of trypsin-cleaved peptides in a mass range of 800–3000 Daltons for HIV-1 gp145 from both cell lines. Furthermore, sequence coverage was similar for gp145 produced in CHO-K1 and Expi293F cells, 53% and 50%, respectively. Collectively, these results confirm that the same protein is being encoded and made by both CHO-K1 and Expi293F cell lines.

**Table 1. Participant characteristics.**

|  | Controls (*n* = 5) | HIV-seropositive (*n* = 15) |
|---|---|---|
| Age (years[1]) | 29 (25, 40) | 39 (24, 58) |
| CD4 cells/mm[3] | - - | 371(18, 1007) |
| HIV RNA copies/mL (log) | - - | 2,63 (1.70, 5.50) 2 ND[2], 2 missing |
| old ART[3] combinations (2006–2012, *n* = 7) | - - | nelfinavir, lamivudine, zidovudine, saquinavir, abacavir, atazanavir |
| new ART[3] combinations (≥2012, *n* = 4) | - - | raltegravir, emtricitabine, tenofovir, etravirine |

[1]median(range),

[2]ND = no detectable,

[3]ART = antiretroviral treatment

## Differences in the glycosylation profiles of CHO-K1- and Expi293F-produced HIV-1 gp145

The distribution of *N*-linked glycans in gp145 produced in CHO-K1 and Expi293F cells was analyzed by MALDI-ToF following PNGase-catalyzed glycan release. The identity of specific glycans was confirmed by post-source decay (PSD) (S2 Fig). MS analysis in reflector positive ion mode revealed a similar distribution of oligomannose-type glycans for gp145 from both cell hosts (Fig 2A and 2C, S1 Table). Specifically, an oligomannose population of $Man_{5-9}GlcNac_2$ with peaks at m/z 1256, 1418, 1580, 1742, and 1904 ($[M + Na]^+$ ions), respectively, was observed. Furthermore, a higher ratio of mannose to complex N-glycans was seen for the protein produced in either cell line. Bi- and tri-antennary complex N-glycans with core fucose (peaks at m/z 1809 and 2174) were observed in both CHO-K1- and Expi293F-produced gp145.

The presence of sialylated N-glycans was determined following hydrolysis of sialic acid residues by neuraminidase A (NEUA). Analysis of the released glycans was performed in both reflector positive and reflector negative ion mode (Figs 2 and 3). The data collected from the

A. HIV-1 gp145-CHO-K1 cells

Sequence Coverage: **53.1**%

B. HIV-1 gp145-Expi293F cells

Sequence Coverage: **49.5**%

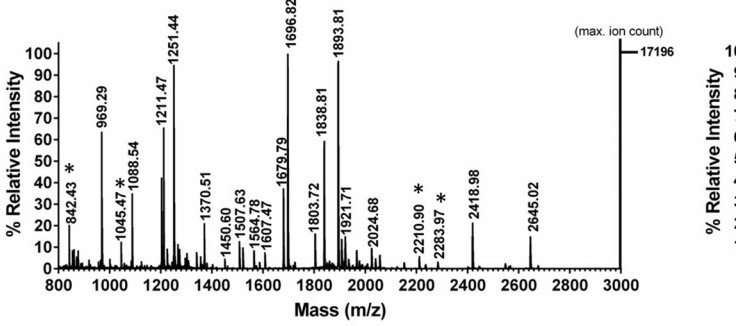
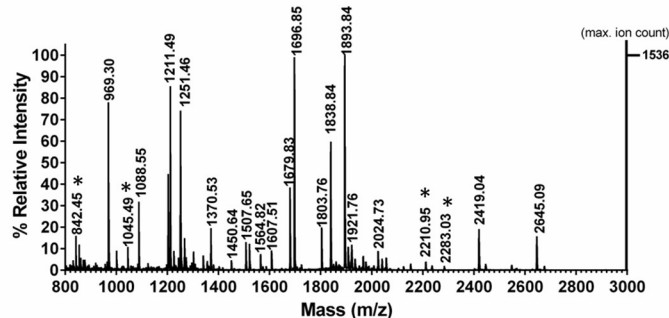

**Fig 1. Peptide mass fingerprinting (PMF) analysis of HIV-1 gp145 protein.** For PMF analysis, 20 μg of gp145 produced in (**A**) CHO-K1 and (**B**) Expi293F cells were used. The gp145 protein was resolved by SDS-PAGE and the 145 kDa band was excised. Gel bands were incubated overnight with trypsin at 37°C, released peptides were co-crystallized with CHCA ionization matrix and the reflector positive mode was used for MALDI-ToF analysis. The x-axis represents the mass-to-charge ratio (m/z) value in Daltons and the y-axis shows the relative abundance (arbitrary units) of the ions. Asterisks (*) highlight the trypsin autolysis peaks.

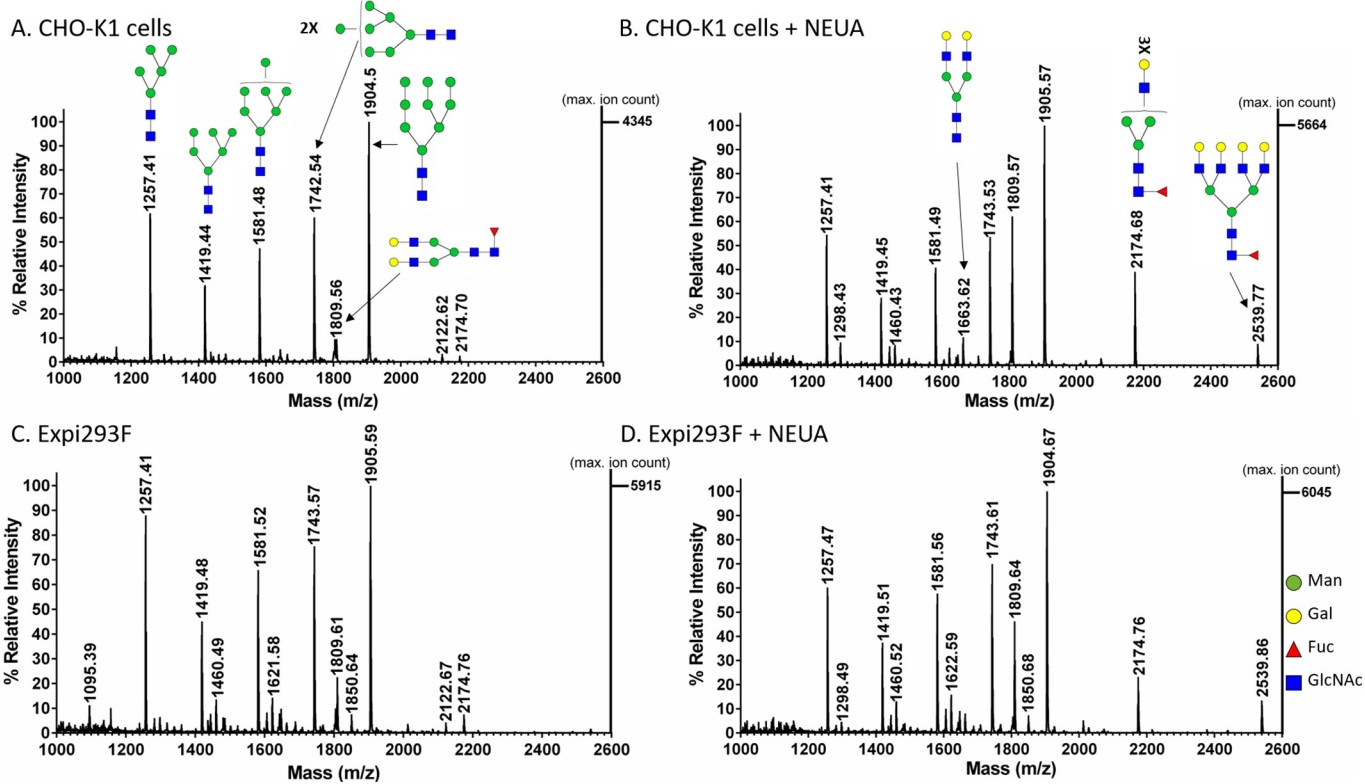

**Fig 2. Positive ion MALDI-ToF analysis of *N*-glycans released from HIV-1 gp145.** The recombinant gp145 (20 μg) produced in (**A-B**) CHO-K1 and (**C-D**) Expi293F cells were incubated overnight with (**B** and **D**) and without (**A** and **C**) Neuraminidase A (NEUA) at 37˚C. Then, gp145 proteins were resolved by SDS-PAGE and the 145 kDa bands were excised. *N*-glycans were released with PNGase F and were co-crystallized with 2',4',6'-Trihydroxyacetophenone monohydrate (THAP) ionization matrix. The x-axis represents the mass-to charge ratio (m/z) value in Daltons and the y-axis shows the relative abundance (arbitrary units) of the ions. The released glycans were analyzed using the MALDI-ToF reflector positive mode. Predicted glycan structures of the masses shown in the upper panel spectrum are the same as those shown in the spectrum on the lower panel.

reflector positive mode revealed the presence of bi-, tri-, and tetra-antennary complex N-glycans with core fucose (peaks at m/z 1808, 2174, and 2538) for gp145 produced in both cell hosts (Fig 2B–2D, S2 Table). A percentage increase in the relative intensity of these glycans in the NEUA-treated samples (in comparison to the untreated samples) suggests that these complex glycans are sialylated. The presence of sialylated glycans was confirmed by the detection of sialylated bi-, tri-, and tetra-antennary complex N-glycans with core fucose (peaks at m/z 1929, 2075, 2390, 2440, and 2754) of the untreated sample in the negative reflector mode (Fig 3A and 3C). When gp145 was treated with NEUA, the signal for these sialylated N-glycans disappeared (Fig 3B and 3D). In order to compare the relative levels of sialylation in CHO-K1- and Expi293F-produced gp145, we plotted the magnitude of neutral glycan increase upon treatment with NEUA for each of the glycan signals (Fig 4). Our results showed that the magnitude of the increase (fold change) is higher for gp145 produced in CHO-K1 cells than for the one produced in Expi 293F. Taken together, these results suggest that CHO-K1 cells produce a gp145 that is more highly sialylated than the one produced in Expi293F.

## Distribution of charge variants

The observed high sialic acid content in the CHO-K1-produced gp145 was expected to result in a lower isoelectric point (pI). To confirm these expected differences in pI, we measured

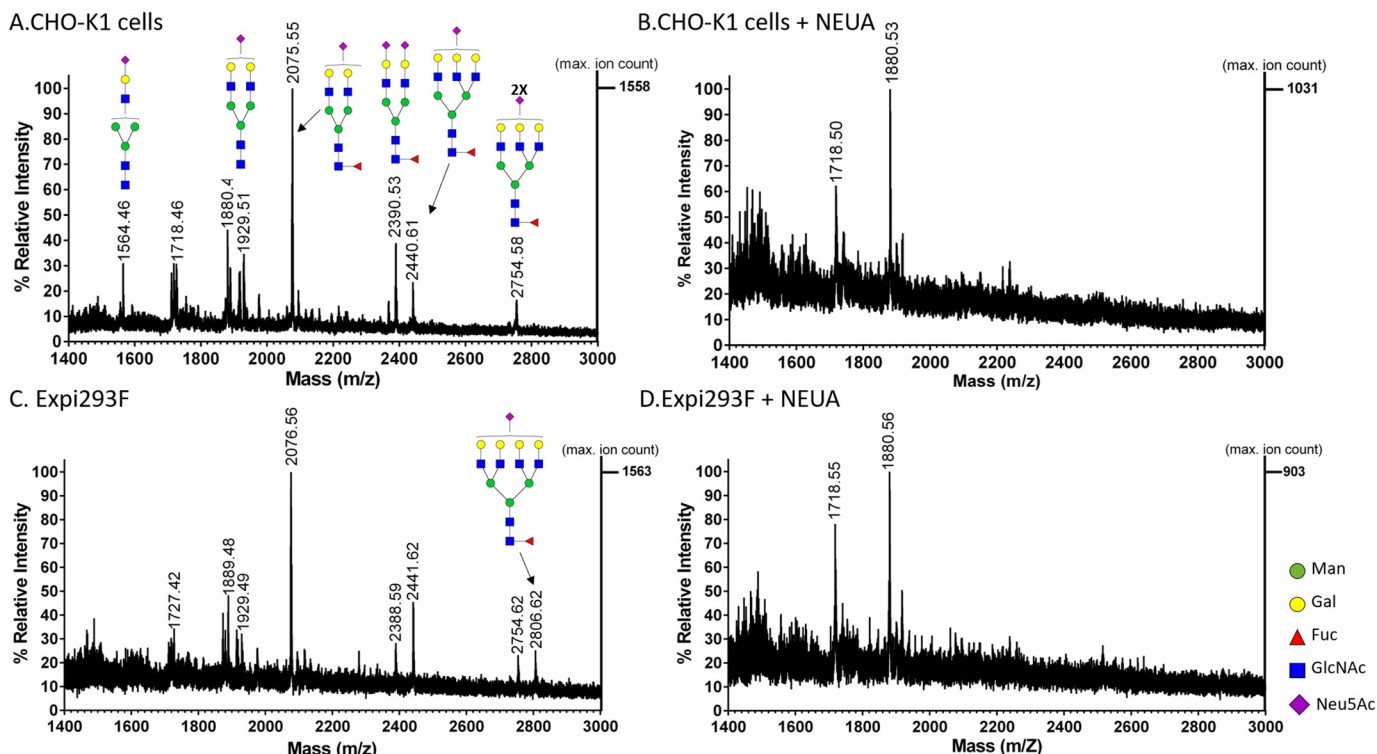

**Fig 3. Negative ion MALDI-ToF analysis of acidic *N*-glycans released from HIV-1 gp145.** The recombinant gp145 (20 μg) produced in (**A-B**) CHO-K1 and (**C-D**) Expi293F cells were incubated overnight with (**B** and **D**) and without (**A** and **C**) Neuraminidase A (NEUA) at 37˚C. Then, gp145 proteins were resolved by SDS-PAGE and the 145 kDa bands were excised. *N*-glycans were released with PNGase F and were co-crystallized with 2',4',6'-Trihydroxyacetophenone monohydrate (THAP) ionization matrix. The x-axis represents the mass-to charge ratio (m/z) value in Daltons and the y-axis shows the relative abundance (arbitrary units) of the ions. The released glycans were analyzed using the MALDI-ToF reflector negative mode. Predicted glycan structures of the masses shown in the upper panel spectrum are the same as those shown in the spectrum on the lower panel.

charge heterogeneity using imaged capillary isoelectric focusing (cIEF) in the presence and absence of NEUA (Fig 5). The pI distribution for the CHO-K1-produced gp145 ranged between 4.3–6.2, whereas that of the Expi293F-produced ranged between 5.4–7.8 (Fig 5A). The observed pI distribution for gp145 produced in both cell hosts was more acidic than the theoretical pI of 8.7 calculated from the amino acid sequence of CO6980v0c22 gp145, however, it is clear that the CHO-K1-produced gp145 is more acidic than the Expi293F-produced gp145. This observation is consistent with a more highly sialylated glycoprotein. As expected, treatment with NEUA shifted the pI of gp145 produced in either cell line to a range of 7.5–9.0 (Fig 5B).

## Recognition of HIV gp145 by the broadly neutralizing antibodies, 2G12 and PG16

Next, we determined the possible effect of host-dependent glycosylation on the recognition of gp145 by broadly neutralizing antibodies (bNAbs), specifically 2G12 [36–40] and PG16 [41]. These antibodies were chosen because their epitope specificities are known to be the high-mannose cluster on the glycan shield and the glycosylation sites in the V1V2V3 loops of the HIV-1 envelope, respectively [42]. Binding was evaluated via biolayer interferometry (BLI) using the FortéBio Octet QK[e] in two different configurations: we either captured the

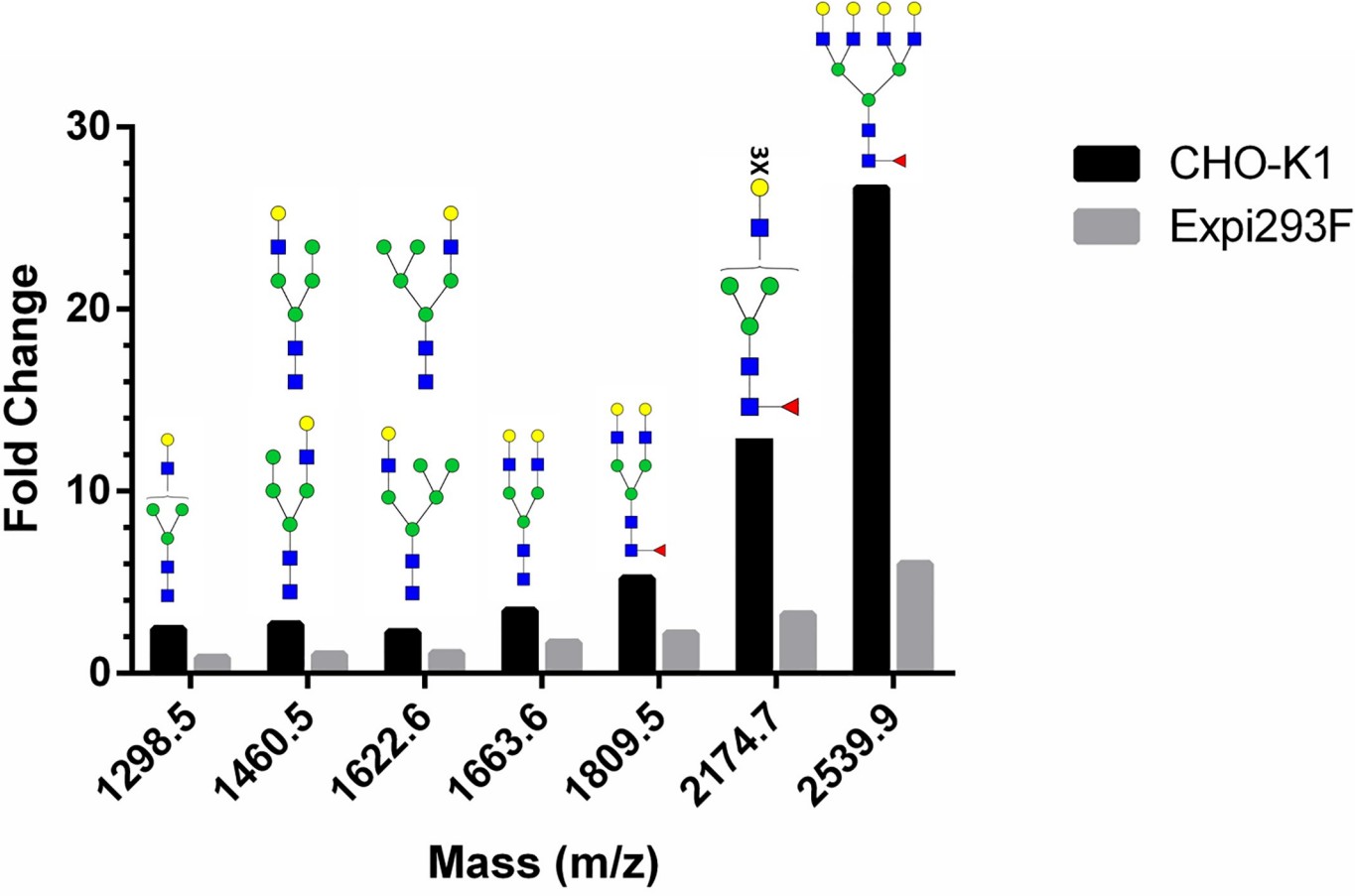

**Fig 4. Abundance of sialylated *N*-glycans of gp145 produced in CHO-K1 and Expi293F cells.** Fold change increase for the composition of acidic *N*-glycans after the addition of Neuraminidase A (NEUA). To calculate the fold change of glycans obtained by MALDI-ToF analysis, the relative abundance of glycans from NEUA-treated samples was normalized against an untreated sample (NEUA+/NEUA-). The relative abundance of a given glycan was determined by an in-house program that adds the ion counts of the user-designated mass and the related isotopic peaks that are plus-or-minus 1, 2, 3 and 4 mass units, then divides the given glycan total by the sum of the ion counts for all glycans under consideration. In this figure, the glycans with the following masses were considered: 1298.5, 1460.5, 1622.6, 1663.6, 1809.5, 2174.7, and 2539.9 (the principal peaks are seen in Fig 2).

corresponding bNAb onto an anti-Fc surface or we immobilized biotin-labeled gp145 on a streptavidin surface.

Using the Fc capture system to immobilize the bNAb, we did not detect any significant differences in binding of CHO-K1-derived and Expi293F-derived gp145 to 2G12 (1.6 nM vs. 1.2 nM; Fig 6A and 6B and Table 2) and PG16 (6.6 nM vs. 2.8 nM; Fig 6C and 6D and Table 2).

However, differences were observed in the binding of 2G12 and PG16 when the CHO- and Expi293F-derived gp145s were immobilized onto the sensor (Table 3). These apparent discrepancies in binding values depending on the assay orientation are not uncommon [43]. When bNAbs are captured on an anti-human Fc sensor, the antibody molecules are oriented in the same fashion exposing the antigen binding region in an optimal and uniform manner. Thus, it is not surprising that higher affinities were obtained when the antibodies were captured, than when the antigen is immobilized in a random orientation [43].

We also measured the binding affinity between CHO-K1- and Expi293F-derived gp145 to other HIV-1 bNAbs and no differences in binding affinity were observed (S3 Table).

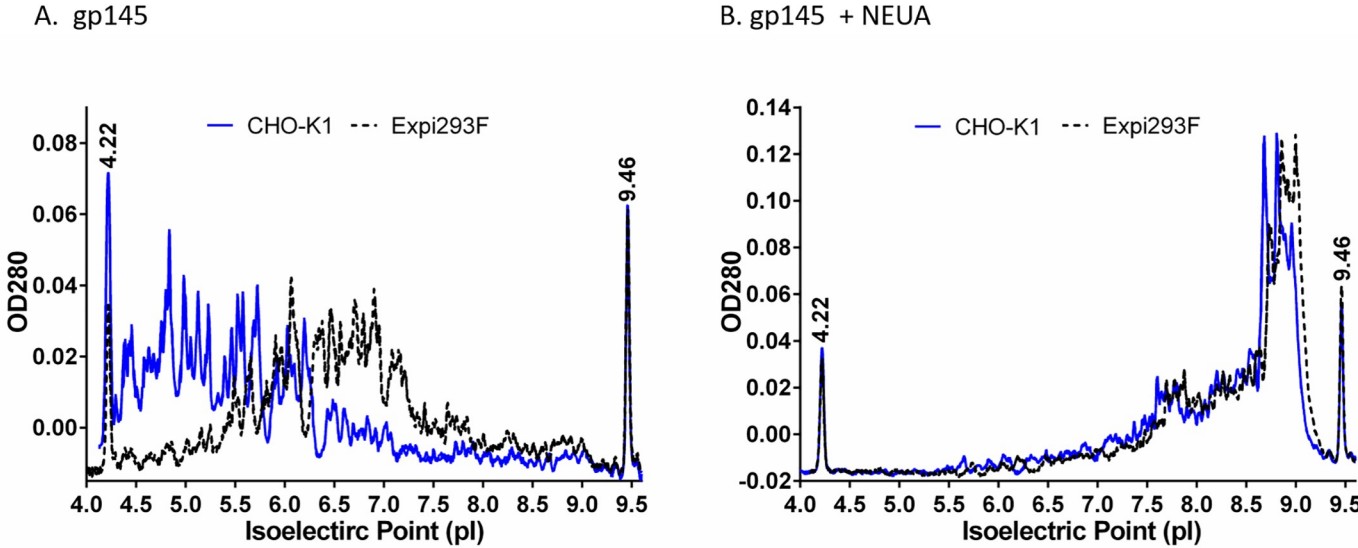

**Fig 5. The impact of sialylation on HIV-1 gp145 isoelectric point.** Electropherograms from 3 consecutive injections of CHO-K1- (blue lines) and Expi293F-produced (black dash lines) gp145 with (**B**) and without (**A**) neuraminidase. The prefocusing was performed for 1 min at 1.5 kV and the focusing for 3.0 kV at 4.5 min. Capillary IEF sample solutions contained 0.7 μg/μL of gp145, 5 M urea, pI markers 4.22/9.46 and servalyte 2–9 carrier ampholytes.

## Reactivity of gp145 against HIV+ and HIV- patient plasma

To assess how differences in glycosylation and charge distribution translate to immunoreactivity, we measured the magnitude of binding antibodies from patient's plasma to gp145 produced in either CHO-K1 or Expi293F cells. Our results showed that the plasma of HIV + individuals reacted significantly over the background against gp145 (Fig 7A), whereas the plasma of uninfected individuals did not react (Fig 7B). Despite the fact that subtype B is the dominant subtype in the Caribbean region [44], plasma from HIV-infected Puerto Ricans showed substantial cross-reactivity with this subtype C immunogen. This cross-reactivity was not surprising since it had been reported by the group of Zolla-Pazner, where they found that individuals infected with clade B virus made antibodies that reacted with Clade C immunogens[45]. Still, no differences in reactivity were observed between gp145 produced in CHO-K1 and Expi293F, indicating that differences in glycosylation do not affect the recognition by HIV infected patient polyclonal antibodies (Fig 7A and 7B).

## Discussion

The HIV envelope glycoprotein (Env) is the most promising candidate for the development of HIV vaccines to date. The short versions of Env, all variants of the gp120, have given mixed results in Phase III efficacy trials, thus highlighting the need for developing new and more complex Env constructs that can elicit a robust immune response. In this paper, we describe efforts to develop a uncleaved gp145 from strain CO6980v0c22, whose amino acid sequence includes the whole gp120 plus a substantial portion of the gp41. This construct was produced using two different expression hosts, and the differences between them were assessed.

The effect of the choice of expression host on the glycosylation and charge distribution for the monomeric gp120 has been explored by others [28,29]. Studies on the glycosylation of A244 and MN gp120 concluded that both proteins are more highly sialylated when expressed in CHO-K1 cells rather than in Expi293F and that this difference in sialylation has some effect

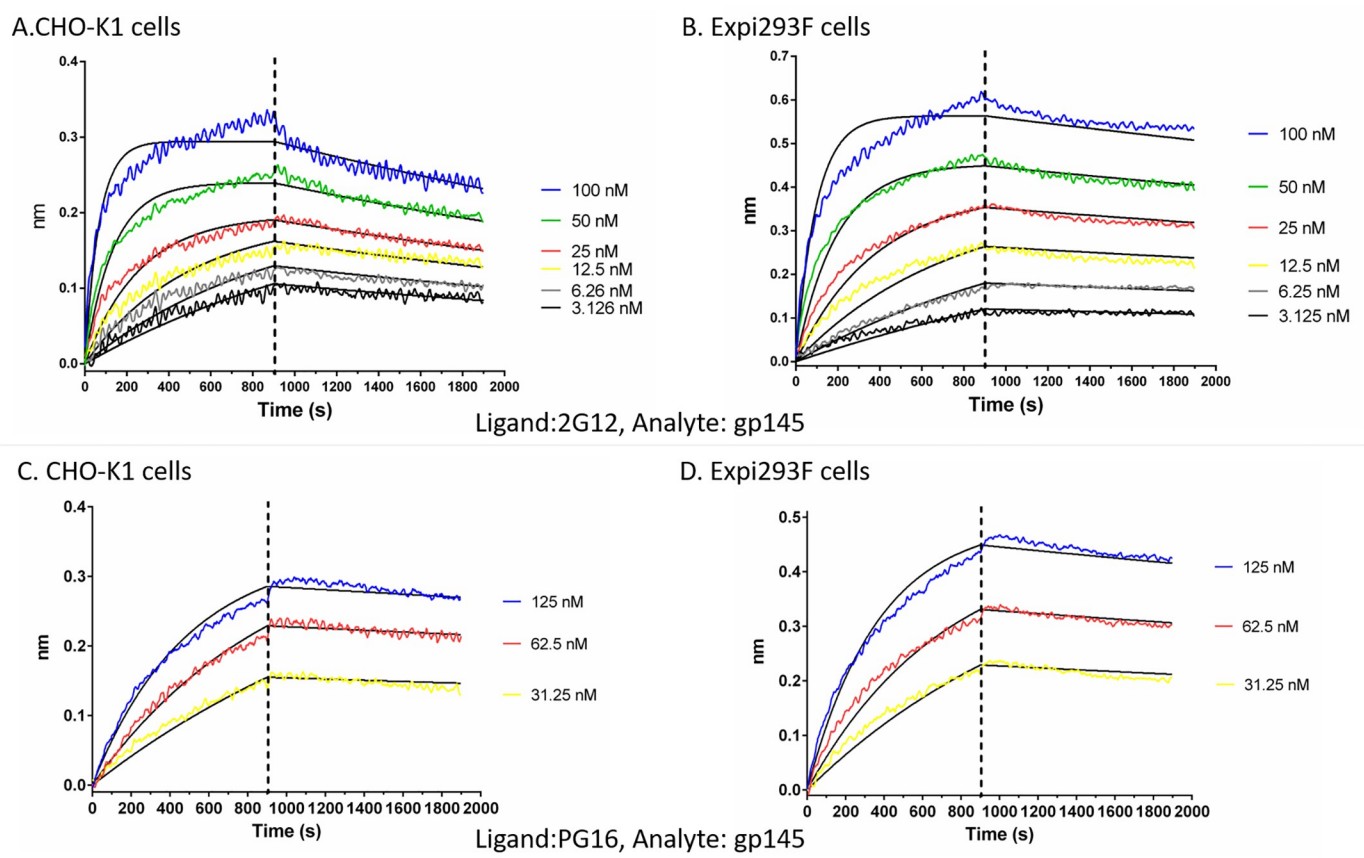

**Fig 6. Binding of 2G12 and PG16 bNAbs to HIV-1 gp145. (A-D)** Octet QK$^e$ sensorgrams generated by binding of serial of dilutions of gp145 produced in CHO-K1 and Expi293F cells to 2G12 and PG16 bNAbs loaded onto anti-human IgG FC capture (AHC) biosensors. Tables show the $k_a$ (rate of association), $k_d$ (rate of dissociation) and $K_D$ (dissociation constant) values for the interactions of HIV-1 gp145 to (**A-B**) 2G12 and (**C-D**) PG16 bNAbs. Black lines represent the 1:1 global fit curve.

on antibody recognition [29]. Similarly, studies on the glycosylation of the consensus B gp120 showed that there are substantial differences in glycosylation depending on the expression host, with Jurkat cells being the host with the most mannose glycosylation [28].

The effect of expression host on glycosylation has been even more closely scrutinized for the production of the native-like SOSIP trimers [30,31]. A detailed study on the glycosylation

**Table 2. Binding of soluble CO6980v0c22 gp145 to Fc-captured bNAbs 2G12 and PG16.**

| Antibody | Expression Host | $k_a$ (1/Ms) | $k_d$ (1/s) | $K_D$ (nM) |
|---|---|---|---|---|
| 2G12 | CHO-K1 | 1.25E+05 | 2.23E-04 | 1.62 ± 0.01 |
| | Expi293F | 9.09E+04 | 1.11E-04 | 1.2 ± 0.2 |
| PG16 | CHO-K1 | 1.67E+04 | 1.06E-04 | 7 ± 3 |
| | Expi293F | 2.09E+04 | 5.97E-05 | 3 ± 1 |

Binding kinetics were obtained using the Octet QKe system. Broadly neutralizing antibodies were captured on an anti-human IgG Fc sensor and dipped into different dilutions of gp145. Association/dissociation rates were analyzed using the ForteBio analysis software. Values represent mean ± standard deviation (STD) from three independent experiments

**Table 3. Binding of bNAbs 2G12 and PG16 to immobilized CO6980v0c22 gp145.**

| Antibody | Expression Host | $k_a$ (1/Ms) | $k_d$ (1/s) | $K_D$ (nM) |
|---|---|---|---|---|
| 2G12 | CHO-K1 | 4.23E+04 | 2.23E-02 | 527 ± 19.3 |
| | Expi293F | 4.52E+05 | 2.98E-02 | 66.0 ± 3.1 |
| PG16 | CHO-K1 | 1.85E+04 | 2.62E-03 | 142 ± 3.7 |
| | Expi293F | 3.26E+04 | 2.05E-03 | 62.8 ± 0.3 |

Binding kinetics were obtained using the Octet Red 96. The CO6980v0c22 gp145 was biotin-labeled and captured on a streptavidin sensor. The resulting sensor was dipped into different dilutions of broadly neutralizing antibodies and association/dissociation rates were analyzed using the ForteBio analysis software. Values represent mean ± KD error.

of BG505.664 SOSIP trimers revealed that, while the gp120 portion of the trimer is mostly decorated with oligomannose glycans, the gp41 portion contains mostly complex glycans, with those made in CHO-K1 cells containing a higher sialic acid content than the cleaved trimers produced in 293 cells [31]. That same study concluded that the uncleaved versions of the glycoprotein contained more complex glycans than the native-like trimeric versions of the corresponding glycoprotein, although this was only reported for gp140 made in 293 cells [31]. Others have reported a similar occurrences of the uncleaved trimer containing a higher proportion of complex glycans [46,47].

In this study, we compared the glycosylation of uncleaved gp145 trimers made in CHO-K1 and Expi293F cells. Our findings that CHO-K1 cells are able to incorporate a larger proportion of sialylated complex glycans is consistent with previous findings [29,31]. These differences in glycosylation did not greatly affect protein recognition by the glycan-specific antibodies, 2G12 and PG16, nor did they affect reactivity against patient-derived antibodies. It was expected that the 2G12 antibody would bind to both CHO- and EXPI293F-produced gp145 with similar affinity, since its epitope is a high-mannose glycan that is likely to be the same in both cell types [48]. Similarly, we did not expect differences in immunogen recognition by PG16 since its epitope has been reported to be

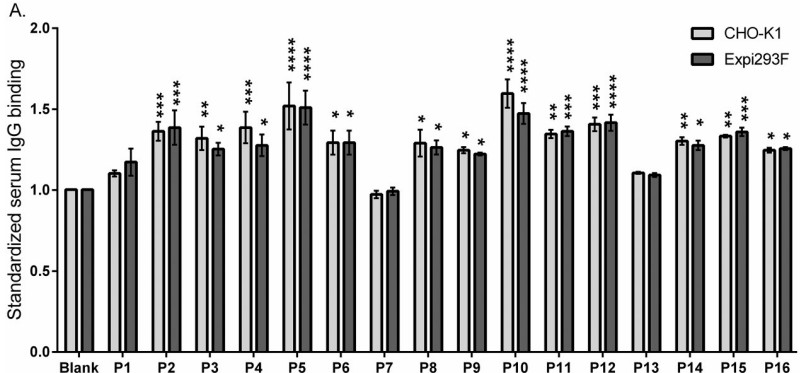
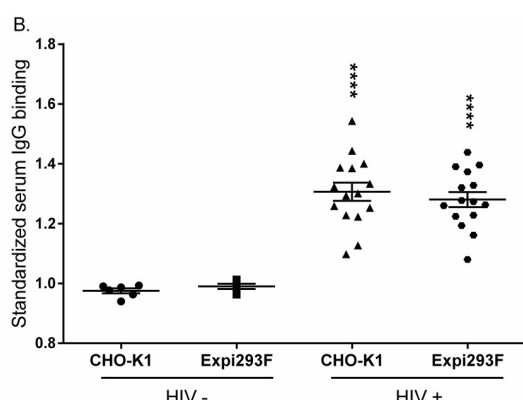

**Fig 7. IgG binding from HIV+ patient plasma to gp145 produced in CHO-K1 and Expi293F cells.** The recombinant protein gp145, produced in either CHO-K1 or Expi293F cells, was tested for its potential to bind antibodies found in HIV+ patients using an ELISA assay. (**A**) Plasma samples from 16 HIV + patients, P1-P16, (1:10,000 dilution) were incubated in 96-well plates coated with gp145 samples produced in two different cell lines (0.5 μg/mL per well). Binding was detected with an anti-human IgG HRP-conjugated antibody. Absorbance was read at 490 nm and standardized against the blank. Values significantly different from the blank (represented by asterisks) were considered reactive. Statistical significance was calculated using a Two-Way ANOVA. *$p<0.05$, **$p<0.01$, ***$p<0.001$, ****$p<0.0001$ (**B**) HIV+ samples were compared with HIV uninfected samples using One-Way ANOVA. Significant differences (represented by asterisks) were found between the two groups but not between immunogens. ****$p<0.001$.

conformational and affected by N-glycosylation, but not sensitive to specific monosaccharide units [49]. The group of Binley reported differences in Env recognition by the glycan-dependent bNAb PG9 that were attributed to the linkage of the sialic acid, which is α-2,6 in 293 cells and α-2,3 in CHO-K1 cells [50]. However, we do not see the same sialic linkage-dependent differences in the binding of PG16, probably a reflection that the epitopes for these two antibodies are overlapping but different.

While there is no clear notion of an optimal glycosylation state for HIV immunogens, recent evidence has revealed that virions contain more complex glycosylation than previously thought [30]. It is generally understood that the sialylation of protein therapeutics extends their pharmacological half-life [51]. However, it is not known whether this increase in the half-life of the glycoprotein would affect vaccine efficacy. Still, the glycosylation of recombinant HIV Env vaccines remains an important feature related to their production yield, quality, and stability. Clearly, the choice of expression host affects glycosylation of the resulting protein in dramatic ways that impact its net charge. Future work should attempt to delineate the relationship between specific glycan content and host immune response. Also, experiments should be performed to identify protein production bottlenecks caused by glycosylation in mammalian cell systems. Overall, glycosylation is an important consideration in the design of new immunogens and in the development of methods for quality analysis.

## Supporting information

**S1 Fig. SDS-PAGE analysis of HIV-1 gp145 produced in CHO-K1 and Expi293F cells.** The recombinant gp145 immunogens were incubated overnight with (+, lanes 3 and 5) and without (-, lanes 2 and 4) Neuraminidase A, and then resolved on 10% SDS-PAGE gels.
(TIF)

**S2 Fig. MALDI-Post-source decay (PSD) spectra of N-linked glycans derived from CHO-K1 gp145.** The presence of Man5GlcNAc2 (A), Man6GlcNAc2 (B), Man7GlcNAc2 (C), Man8GlcNAc2 (D), Gal2Man3GlcNAc4Fuc1 (E) and Man9GlcNAc2 (F) was confirmed by MALDI-PSD. The x-axis represents the mass-to charge ratio (m/z) value in Daltons and the y-axis shows the relative abundance (arbitrary units) of the ions.
(TIF)

**S1 Table. *N*-glycans composition of C06980v0c22 gp145 using MALDI-ToF in reflector positive ion mode.** All the peaks obtained in the MALDI-ToF were compared against the list previously reported by Doores et al., 2010. Some of them were selected randomly for confirmation by MS/MS analysis.
(DOCX)

**S2 Table. N-glycans composition of C06980v0c22 gp145 using MALDI-ToF in reflector negative ion mode.**
(DOCX)

**S3 Table. Binding of CO6980v0c22 gp145 to HIV-1 antibodies.** Binding kinetic results were obtained using the Octet Red 96 system. Biotinylated gp145 was immobilized on streptavidin sensors, dipped into two-fold dilutions of the monoclonal antibodies and association/dissociation rates were analyzed using the Octet Molecular Interaction System software. *All CHO-K1 antigenicity results were previously reported by Wieczorek *et al.* 2015 (16).
(DOCX)

## Acknowledgments

The authors also thank Advanced Biosciences Laboratories (ABL), Inc. for providing the CO6980v0c22 gp145 reference material and Ms. Elaine Rodríguez for technical support and sample retrieval. A special note of thanks goes to Ms. Nandini Sane from the NIH Division of AIDS providing critical feedback. Also, the authors acknowledge WHO-UNAIDS and the NIH AIDS Reagent Program for providing bNAbs for the current study. Specifically, 1) the VRC01 reagent was obtained through the NIH AIDS Reagent Program, Division of AIDS, NIAID, NIH: Anti-HIV-1 gp120 Monoclonal (VRC01), from Dr. John Mascola (cat# 12033); 2) the 2G12 reagent was obtained through the NIH AIDS Reagent Program, Division of AIDS, NIAID, NIH: Anti-HIV-1 gp120 Monoclonal (2G12) from Polymun Scientific; 3) the PG16 reagent was obtained through the NIH AIDS Reagent Program, Division of AIDS, NIAID, NIH: Anti-HIV-1 gp120 Monoclonal (PG16) from IAVI.

## Author Contributions

**Conceptualization:** José A. González-Feliciano, Pearl Akamine, Coral M. Capó-Vélez, Manuel Delgado-Vélez, Shelly J. Krebs, Abel Baerga-Ortiz.

**Data curation:** José A. González-Feliciano, Pearl Akamine, Coral M. Capó-Vélez, Vincent Dussupt, Shelly J. Krebs, Victoria R. Polonis, Abel Baerga-Ortiz.

**Formal analysis:** José A. González-Feliciano, Pearl Akamine, Manuel Delgado-Vélez, Vincent Dussupt, Valerie Wojna.

**Funding acquisition:** Manuel Delgado-Vélez, José A. Lasalde-Dominicci.

**Investigation:** José A. González-Feliciano, Vincent Dussupt, Shelly J. Krebs, Victoria R. Polonis.

**Methodology:** Vincent Dussupt, Valerie Wojna.

**Project administration:** Manuel Delgado-Vélez, Abel Baerga-Ortiz.

**Resources:** Shelly J. Krebs, Valerie Wojna, Victoria R. Polonis.

**Software:** José A. Lasalde-Dominicci.

**Supervision:** Manuel Delgado-Vélez, Victoria R. Polonis, Abel Baerga-Ortiz, José A. Lasalde-Dominicci.

**Validation:** Valerie Wojna.

**Writing – original draft:** José A. González-Feliciano, Pearl Akamine, Coral M. Capó-Vélez, Victoria R. Polonis, Abel Baerga-Ortiz.

**Writing – review & editing:** Coral M. Capó-Vélez, Manuel Delgado-Vélez, Vincent Dussupt, Shelly J. Krebs, Valerie Wojna, Victoria R. Polonis, Abel Baerga-Ortiz, José A. Lasalde-Dominicci.

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
