## [Decision Letter · Decision Letter 0]

14 Apr 2020

PONE-D-20-07724

A recombinant gp145 Env glycoprotein from HIV-1 expressed in two different cell lines: effects on glycosylation and antigenicity

PLOS ONE

Dear Dr. Baerga-Ortiz,

Thank you for submitting your manuscript to PLOS ONE. After careful consideration, we feel that it has merit but does not fully meet PLOS ONE’s publication criteria as it currently stands. Therefore, we invite you to submit a revised version of the manuscript that addresses the points raised during the review process.

Both Reviewers agreed regarding the good technically quality of your submitted manuscript.

However, they have risen a couple of major concerns,

First why did you choose to test gp145 antigenicity using clinical plasma samples from a population mostly infected with an heterologous clade.

In second place, why did you pick uncleaved gp145 as immunogen candidate to be scaled up, considering other current designs/approaches that may work better.

Please, read the detailed comments from each Reviewer below.

We would appreciate receiving your revised manuscript by May 29 2020 11:59PM. To enhance the reproducibility of your results, we recommend that if applicable you deposit your laboratory protocols in protocols.io, where a protocol can be assigned its own identifier (DOI) such that it can be cited independently in the future. For instructions see: http://journals.plos.org/plosone/s/submission-guidelines#loc-laboratory-protocols

We look forward to receiving your revised manuscript.

Kind regards,

Juan Pablo Jaworski, D.V.M., M.Sc., Ph.D.

Academic Editor

PLOS ONE

Journal Requirements:

Reviewers' comments:

Reviewer's Responses to Questions

**Comments to the Author**

1. Is the manuscript technically sound, and do the data support the conclusions?

Reviewer #1: Yes

Reviewer #2: Yes

2. Has the statistical analysis been performed appropriately and rigorously? 

Reviewer #1: No

Reviewer #2: Yes

3. Have the authors made all data underlying the findings in their manuscript fully available?

Reviewer #1: Yes

Reviewer #2: Yes

4. Is the manuscript presented in an intelligible fashion and written in standard English?

Reviewer #1: Yes

Reviewer #2: Yes

5. Review Comments to the Author

Reviewer #1: In the manuscript presented by Gonzalez-Feliciano et al, the authors study the glycosylation pattern of the HIV surface protein gp145 in its uncleaved form utilizing two mammalian cell lines, CHO-K1 and Expi293F. The antigen is the primary target for vaccine development against HIV. The authors present literature supporting the notion that different expression systems differ in the way gp145 is decorated, having a potential impact in the generation of and recognition by bNAb. The authors found relevant differences regarding the nature of the glycans in gp145 produced in the two cells lines using suitable technology. They also assess the impact on antigenicity given by glycan differences using bnMAb and polyclonal sera from infected patients. Overall this is an interesting and exciting work that is well written and the subject is of relevance to the field. However, in this reviewer's modest opinion, the manuscript needs minor and major revisions to be considered for publication.

Minor revisions

Figure 2C. Plot shows peak at 1742. The number above the peak reads 17423.57. Please fix the decimal point.

Figures 1, 2, 3 and 4. Please clarify what unit m/z is.

Populations?

Font/size of figure 3 B differ from the other panels in the figure.

Background noise figure 3B and D?

In the figures 3 C there is a peak at ~2800 that is not identified that has a relative intensity comparable with some that are identify and is a potential difference with 3 A. Could you please provide details on why this can be omitted?

Line 288. Please define unit “au”.

Line 289-291. Here you refer to the peaks that you included in the analysis. I) the figure show 7 peaks that were analyzed whereas in the text you mentioned 12. I would like to know why is this difference. Also, the numbers mentioned in the text do not match the number in the figure 2. For example, in figure 2 the peak reads as 1257.4, in all four panels, but in the text you refer to this peak as 1256. Please, these discrepancies need to be fixed.

Line 290. “12987.5” I think is 1298.75.

Line 295. You first refer to isoelectric point and then pI. I suggest to add “(pI)” after “isoelectric point” in this line.

Comparison of bNAb. Here you show that immobilization of the antibody does not show differences in the recognition of gp145 produced by any of the cell lines. Whereas when you use gp145 attached to the surface you do see differences. You justify these differences by the way the antigen is presented to the antibody. Even though I do believe these data need to be published, I would suggest that is shown as supplementary data, so it does not affect the interpretation of the of the other, more solid, result.

Line 358-359: “…even when stratifying for antiretroviral treatment status.” I do not find this stratification anywhere in the figure 7 or anywhere else in the manuscript. Please, either add the missing data or remove this statement.

Figure 7. Please, add what is the meaning of the asterixis in the figure.

I would like if you could add to the discussion what would be the importance of the protein presenting more glycans bearing sialic acid. You used approaches specifically to differentiate these from other glycans but do not mention why they are important in this context neither discuss it. Please, I suggest that you add this information since it would help to put the importance of these differences in context.

Major

In the analysis of the antibody binding you make significant inferences from the data without running any statistical method. I think it is important to show by some statistical method that these differences in binding are significant or not and then based on that discuss the results.

Line 354-355 “Despite the fact that subtype B is the dominant subtype in the Caribbean region…” The sera used to assess reactivity (figure 7), were all from patients infected with subtype B? Or since this is the main subtype in the Caribbean this is an assumption? If available, this information needs to be added to the manuscript. In any way, I would like to see some deeper discussion on how this may affect your results. If there is more literature on this, please cite and discuss. As you acknowledge, subtype mismatch may affect the way these sera recognize the antigen. In other viruses, these mismatches between subtypes may go from none recognition to partial or fairly good recognition. I guess what you are showing belongs to the later. Do you think that heterologous sera may recognize better or worse subtle differences in the glycosylation of gp145?

Reviewer #2: This paper compares the antigenicity and glycan profiles of a clade C gp145 expressed either in CHO-K1 and Expi cells. The authors found the gp145’s to differ in isoelectric point and sialic acid incorporation, but that this did not translate into major antigenic differences in terms of 2G12, PG16 or HIV+ plasma binding. They conclude that there are sizeable differences in glycosylation depending on host cell, but that it is unclear how these differences will translate into vaccine efficacy.

The paper is technically really good. However, the question comes with the samples. It’s not clear why the clade C strain was chosen. Nor is it clear why it was generated as an uncleaved gp145, especially considering the now substantial evidence for better folded forms of trimer ectodomain that would be more authentic representations of the surface spikes from the clade C virus they chose. Uncleaved gp140 is not compact and glycans tend to be more processed compared to native spikes, because of the fewer constraints on glycan processing. It would be even better if they were truly native trimers, as expressed in membranes. The use of a non-native form of gp140 inevitably reduces the power of the findings.

The authors initially did not find any notable differences in binding by 2G12 and PG16. However, in Table 3, line 334, they observed “differences” (lower binding?) without explaining what they mean. For example, do they normalize PG16 binding to that of 2G12? Or do they compare the patterns in the two orientations? Or do they mean that the capture methods lead to different outcomes that may not reflect antibody binding differences? For 2G12, this is not surprising since this antibody recognizes an invariant high mannose epitope that is unlikely to be affected by producer cells. For PG16, the lack of change may in part trace to the fact that CHO cells tend to add alpha 2,3 sialic acids, whereas PG16 prefers the 2,6 sialic acids more commonly found in human production (PMID: 29718999), so PG16 is ultimately ambivalent or even slightly averse to the CHO cell 2,3 glycans. There is in fact a slight loss of PG16 binding to CHO cells, as compared to the control using 2G12 as a control arbiter for binding. Overall, I am not sure the data in S3 Table and elsewhere (Table 2, 3 and other kinetic data) for the Octet work are not different for the two gp145’s. A lot of tabulated Kds could be plotted to check for antibody-specific patterns. KD’s are not massively different, but the degree of difference seems to vary per antibody which may be worth capturing. Otherwise this would not fully investigate the patterns that are justified by the effort in running all these affinity tests.

The two gp145 preps were also purified differently. It is unclear what the method for the CHO version was as it refers to another paper (a brief description would help). A major question is whether these gp145s are monomers or oligomeric forms or various? These different forms will bear different glycans so this is another problem leading to variability, in addition to using uncleaved gp145.

line 70: Native trimers here are actually “near native” and should be referred to as such. While they are a closer match to native, membrane trimers than uncleaved gp145, there are several differences.

line 79: Re: a tremendous strain on the protein trafficking machinery, the other side of this point is that the glycans play a key structural role in folding. Removing some glycans can decrease expression of trimers. So there is more of a trade off with glycans being present.

line 92: There have been some prior analyses on uncleaved gp140. A google search of “uncleaved gp140 glycans” revealed a few articles, including (PMID: 26051934, PMID: 26018173), and there have been a few by Go/Desaire. Uncleaved gp140 glycans tend to be more processed according to (PMID: 26051934), consistent with their less compact, non-native conformation.

line 311: The subtitle heading would be better reversed to show that it is the antibodies binding to the gp145 and not the other way around.

6. PLOS authors have the option to publish the peer review history of their article (what does this mean?). If published, this will include your full peer review and any attached files.

Reviewer #1: Yes: Lucas M. Ferreri

Reviewer #2: No

---

## [Author Response · Author response to Decision Letter 0]

2 May 2020

Response to Reviewer #1:

Thank you for taking the time to read our manuscript thoroughly and for pointing out the technical suitability of the biochemical methods employed to address the question of glycan composition and antigenicity. We have accepted all of your minor revisions and have amended the manuscript to reflect that. Below is a point-by-point response to your comments: 

“Minor revisions

Figure 2C. Plot shows peak at 1742. The number above the peak reads 17423.57. Please fix the decimal point.”

We have fixed this mistake in Figure 2C . Thank you for pointing it out.

“Figures 1, 2, 3 and 4. Please clarify what unit m/z is.”

The legends now include an explanation of what the x- and y- axes represent. The lableing of the x-axis of MALDI-ToF spectra as “m/z” is consistent with the published literature of the field. 

“Font/size of figure 3 B differ from the other panels in the figure.”

The figure has been amended.

“Background noise figure 3B and D?”

Figure 3 present the MALDI-ToF spectra in the negative mode for glycans that were either fully sialylated (3A and 3C) or enzymatically desialylated (3B and 3D). Because desialylated glycans lose their negative charge their signal is dampened in the negative mode, hence the apparent increase in the noise. 

“In the figures 3 C there is a peak at ~2800 that is not identified that has a relative intensity comparable with some that are identify and is a potential difference with 3 A. Could you please provide details on why this can be omitted?”

Thank you for pointing this out. The 2806 signal corresponds to the tetra-antenary sialylated glycans. We have now identified the signal in figure 3C. The signal for this glycan is not one that we detect consistently. Thus, we would not be comfortable drawing conclusions from the presence or absence of this speficic signal at in the current data.

“Line 288. Please define unit “au”.”

Has been changed to “mass units” and it is now in Line 295

“Line 289-291. Here you refer to the peaks that you included in the analysis. I) the figure show 7 peaks that were analyzed whereas in the text you mentioned 12. I would like to know why is this difference. Also, the numbers mentioned in the text do not match the number in the figure 2. For example, in figure 2 the peak reads as 1257.4, in all four panels, but in the text you refer to this peak as 1256. Please, these discrepancies need to be fixed.”

The figures have been changed and the text has been amended.

“Line 290. “12987.5” I think is 1298.75.”

The text has been amended and it is now Line 295

“Line 295. You first refer to isoelectric point and then pI. I suggest to add “(pI)” after “isoelectric point” in this line.”

The text has been amended.

“Comparison of bNAb. Here you show that immobilization of the antibody does not show differences in the recognition of gp145 produced by any of the cell lines. Whereas when you use gp145 attached to the surface you do see differences. You justify these differences by the way the antigen is presented to the antibody. Even though I do believe these data need to be published, I would suggest that is shown as supplementary data, so it does not affect the interpretation of the of the other, more solid, result.”

The reviewer rightly points out the major discrepancy between the two methods of immobilizing molecules for a biosensor binding study. When the antibody is captured on an anti-Fc sensor, all of the antibody molecules are captured in the same orientation. Thus, the resulting sensor surface is fully competent and the data resulting from it should be more reliable. Thus, our interpretation and conclusion that differences in glycosylation and more specifically, sialyation, do not greatly impact antibody binding, are based on the measurements obtained with the captured antibody. We had many discussions internally on whether the binding data obtained with the covalently immobilized gp145 would be more appropriately placed in the supplementary section, but we did not want it to be perceived like we were hiding conflicting results. However, we would be happy to move that table 3 and binding data from the immobilized gp145 to the supplementary section, if that is the reviewer’s recommendation.

“Line 358-359: “…even when stratifying for antiretroviral treatment status.” I do not find this stratification anywhere in the figure 7 or anywhere else in the manuscript. Please, either add the missing data or remove this statement.”

The phrase “…even when stratifying for antiretroviral treatment status.” Has been deleted and is now in Line 368.

“Figure 7. Please, add what is the meaning of the asterixis in the figure.”

We have changed the figure legen to included the statistical test applied to determine significance, and the p-value assigned to each group of asterisks in Lines 376-382.

“I would like if you could add to the discussion what would be the importance of the protein presenting more glycans bearing sialic acid. You used approaches specifically to differentiate these from other glycans but do not mention why they are important in this context neither discuss it. Please, I suggest that you add this information since it would help to put the importance of these differences in context.”

In the world of injectable protein therapeutics a higher proportion of sialic acid is generally thought to contribute to a longer pharmacological half-life once injected, which is typically desirable for most protein therapeutics. This paradigm could be different in the context of vaccines, since a longer half-life is not necessarily correlated with immunogenicity. Thus, the impact of glycosylation on the eventual efficacy of an immunogen is an interesting question that remains somewhat unexplored. We have added a sentence and a new reference (Werner et al., 2007). We have added a sentence on this topic in Lines 429-431.

Major revisions

“In the analysis of the antibody binding you make significant inferences from the data without running any statistical method. I think it is important to show by some statistical method that these differences in binding are significant or not and then based on that discuss the results.”

The following line was added to the figure legend in Lines 378-382: 

“Statistical significance was calculated using a Two-Way ANOVA. *p<0.05, **p<0.01, ***p<0.001, ****p<0.0001. HIV+ samples were compared with HIV uninfected samples using One-Way ANOVA. Significant differences (represented by asterisks) were found between the two groups but not between immunogens. ****p<0.001”

“Line 354-355 “Despite the fact that subtype B is the dominant subtype in the Caribbean region…” The sera used to assess reactivity (figure 7), were all from patients infected with subtype B? Or since this is the main subtype in the Caribbean this is an assumption? If available, this information needs to be added to the manuscript.”

Thank you for the question, since this point was not made clear in the manuscript. The predominant subtype in Puerto Rico (and the rest of the Caribbean) is subtype B, and therefore, we assume that all of our patient sera analyzed are subtype B, although we cannot say that with any measure of scientific certainty. A change has been made in the text to reflect the fact that we do not know the subtype of HIV present in patient serum. Some lines of text have been added to the results section in Lines 363-365.

“In any way, I would like to see some deeper discussion on how this may affect your results. If there is more literature on this, please cite and discuss. As you acknowledge, subtype mismatch may affect the way these sera recognize the antigen. In other viruses, these mismatches between subtypes may go from none recognition to partial or fairly good recognition. I guess what you are showing belongs to the later. Do you think that heterologous sera may recognize better or worse subtle differences in the glycosylation of gp145?”

Some years ago, the group of Susan Zolla-Pazner at NYU examined the sera of both individuals infected with HIV subtype B and immunized with Env from a subtype B strain (Verrier et al. 2000). The IgG’s from both of these groups exposed to subtype B, were found to bind to peptides and recombinant Env’s from different clades. They especially cross-reacted with clade C peptides. Thus, it was not entirely surprising to see this robust cross-clade reactivity in our patient serum samples. We have included a sentence on this previous work by Zolla-Pazner in Lines 363-365.

Although this has not been thoroughly explored, we expected that at least some of the cross-reactive IgGs would recognize the glycans and we also expected that the difference in total negative charge between the two proteins would be large enough to affect IgG recognition. However, we did not see that. IgGs from Puerto Rico patients bound to both gp145 made in CHO (highly sialylated and negative) and gp145 made in 293 (less sialylated and less negative).

Response to Reviewer #2:

Thank you for taking the time to read our manuscript thoroughly and for providing good explanations for our antibody binding results. We appreciate your observation of the technical suitability of the biochemical methods employed in our work to address the question of glycan composition and antigenicity. Below is a point-by-point response to your comments: 

“It’s not clear why the clade C strain was chosen. Nor is it clear why it was generated as an uncleaved gp145, especially considering the now substantial evidence for better folded forms of trimer ectodomain that would be more authentic representations of the surface spikes from the clade C virus they chose. Uncleaved gp140 is not compact and glycans tend to be more processed compared to native spikes, because of the fewer constraints on glycan processing. It would be even better if they were truly native trimers, as expressed in membranes. The use of a non-native form of gp140 inevitably reduces the power of the findings.”

The reviewer rightly notices that although HIV infections in Puerto Rico and the rest of the Caribbean are of clade B strains, our group is leading efforts to produce a vaccine based on a clade C strain. The production of clade C vaccine was an assignment from the NIH under grant R01AI122935, the aims of which included the optimization of titer and yield for a CHO cell line expressing the CO6980v0c22 gp145, originally identified in a Tanzanian strain. This immunogen had previously shown to be notoriously difficult to produce in large amounts with diminishing returns on scale-up. Our group has been able to increase the scale and the titers for this problematic construct through the systematic optimization of media conditions. We understand that the NIH is leading efforts to produce numerous vaccines of different clades and CO6980v0c22 happens to be the one that we were assigned or encouraged to work with.

We understand the reviewer’s comment on developing Env versions that are considered to be more physiological or native-like than the uncleaved trimers. However, it is difficult to predict what an effective and protective HIV vaccine will look like. While it is true that the SOSIP trimers have been successfully built to closely resemble the native trimers, their performance in Phase I clinical trials has not yet been reported. They could very well perform as designed, but this remains to be tested. At the present time, there are numerous other types of cleaved, uncleaved and single-domain constructs that are very much still under consideration and undergoing Phase I vaccine trials. The NFL trimers have shown good promise and so have vaccines templated on the external outer domain (eOD) of gp120 and on the fusion peptide (FP) of gp41.

Currently, one of the most anticipated clinical trials (NCT03060629), which is sponsored by Janssen Vaccines, involves the immunization of human participants with the Mosaic uncleaved gp140 trimer as a boost immunogen. While we do not know the results from this Phase 2b trial, it is fair to say that uncleaved trimers are very much under consideration and our work is a step toward understanding glycosylation and sialylation in this family of Env constructs. 

Our CO6980v0c22 gp145 is slightly longer than the Mosaic gp140 as it includes the MPER region which is a known epitope for broad neutralization. This construct has been shown to form trimers and also dimers, in a similar proportion as Env constructs that are considered to be more physiological (Wieczorek et al., 2015). Furthermore, the CO6980v0c22 gp145 is currently undergoing Phase I clinical trials (NCT03382418) for safety and immunogenicity and the trial results should become available soon. 

While we agree with the reviewer that there are multiple HIV Env versions out there, it is still too early to discard or dismiss any of them, especially while they are undergoing clinical testing.

“The authors initially did not find any notable differences in binding by 2G12 and PG16. However, in Table 3, line 334, they observed “differences” (lower binding?) without explaining what they mean. For example, do they normalize PG16 binding to that of 2G12? Or do they compare the patterns in the two orientations? Or do they mean that the capture methods lead to different outcomes that may not reflect antibody binding differences?”

The last question posed by the reviewer reflects our thinking. When antibodies are uniformly captured on the sensor, we do not see differences in binding between CHO and 293 cells. But when the gp145 is covalently attached via lysine amides, then we see differences in binding. These differences could be due to the mode attachment. In covalent linkage, the proteins are coupled onto a negatively charged surface. Since the gp145 CHO-K1 is a more negative protein than the gp145 made in 293, it will attach differently via the covalent method. Thus, it is our thinking that the data obtained with the sensor generated via antibody capture is, at least in this case, more reliable.

“For 2G12, this is not surprising since this antibody recognizes an invariant high mannose epitope that is unlikely to be affected by producer cells. For PG16, the lack of change may in part trace to the fact that CHO cells tend to add alpha 2,3 sialic acids, whereas PG16 prefers the 2,6 sialic acids more commonly found in human production (PMID: 29718999), so PG16 is ultimately ambivalent or even slightly averse to the CHO cell 2,3 glycans. There is in fact a slight loss of PG16 binding to CHO cells, as compared to the control using 2G12 as a control arbiter for binding.”

We were unaware of this preference of PG16 for the 2,6-linked sialic acid and its selection against the 2,3-linked sialic acid. Our discussion was based on the findings of Doores and Burton from 2010 (PMID 20686044) that both PG9 and PG16 recognize overlapping conformational epitopes that are affected by N-glycosylation perhaps indirectly, but are not sensitive to specific monosaccharide units. However, a more recent report by Crooks et al., 2018 concludes that the binding of PG9 could be enhanced by hypersialylation in human cells. No mention is made in that report about bNAb PG16. A note has been added to the discussion in Lines 418-426.

“Overall, I am not sure the data in S3 Table and elsewhere (Table 2, 3 and other kinetic data) for the Octet work are not different for the two gp145’s. A lot of tabulated Kds could be plotted to check for antibody-specific patterns. KD’s are not massively different, but the degree of difference seems to vary per antibody which may be worth capturing. Otherwise this would not fully investigate the patterns that are justified by the effort in running all these affinity tests.”

We totally agree with this interpretation of the binding data. We do not think that the binding of glycan-specific antibodies is any different for both of these immunogens, despite their differences in complex glycosylation.

“The two gp145 preps were also purified differently. It is unclear what the method for the CHO version was as it refers to another paper (a brief description would help). A major question is whether these gp145s are monomers or oligomeric forms or various? These different forms will bear different glycans so this is another problem leading to variability, in addition to using uncleaved gp145.”

Both immunogens were purified through a GNL lectin affinity column followed by Q-sepharose chromatography, which resulted in the production of a mixture of oligomeric forms that was not further resolved. We have now included a brief description of the purification of gp145 from CHO-K1 in the Methods section in Lines 108-112.

Regarding the possible differences in glycosylation for each oligomeric form, in the past our group has analyzed the glycosylation of trimers compared with monomers of different Env constructs, and have observed no differences. These analyses have been done as part of contract work for a manufacturing organization and we have not received permission to publish that data. Our explanation for this lack of difference in glycosylation between trimers and monomers is that the sub-units are in dynamic equilibrium between the different oligomeric states, causing a homogenization of the glycosylated forms. 

“line 70: Native trimers here are actually “near native” and should be referred to as such. While they are a closer match to native, membrane trimers than uncleaved gp145, there are several differences.”

We have substituted the word “native” with “native-like”

“line 79: Re: a tremendous strain on the protein trafficking machinery, the other side of this point is that the glycans play a key structural role in folding. Removing some glycans can decrease expression of trimers. So there is more of a trade off with glycans being present.”

Thank you for pointing this out. While extensive glycosylation poses challenges to the efficient production of glycoproteins, it is essential for protein function. It has been shown that the complete removal of Env glycans from pseudovirions eliminates infectivity (Binley et al., 2010; Ref #17 in our manuscript). Thus glycans are important for function and thus must be present on any vaccine candidate that resembles the infective virus. 

We have followed the statement quoted by the reviewer with the following statements: “It has been generally understood that for an Env vaccine to be effective, its glycan shield should resemble that of the infectious virus. Thus, a number of analytical strategies have been developed to measure the precise chemical nature and distribution of the glycans in Env from viral and recombinant sources.” in Lines 80-83.

“line 92: There have been some prior analyses on uncleaved gp140. A google search of “uncleaved gp140 glycans” revealed a few articles, including (PMID: 26051934, PMID: 26018173), and there have been a few by Go/Desaire. Uncleaved gp140 glycans tend to be more processed according to (PMID: 26051934), consistent with their less compact, non-native conformation.”

We have acknowledged prior work on the glycosylation of uncleaved gp140 vs. their native-like version, in which authors showed that uncleaved gp140 is decorated with a higher proportion of complex glycans than the native-like (Pritchard et al., 2015; PMID 26051934), which the reviewer has also identified. We have now added additional references suggested by the reviewer in support of the observation that uncleaved trimers contain more mannose glycans compared to their SOSIP or their membrane-embedded versions, respectively (Ringe et al., 2015 PMID 26311893 and Go et al., 2015 PMID 26018173). They are cited in Lines 412-413. Thank you for the suggestion.

“line 311: The subtitle heading would be better reversed to show that it is the antibodies binding to the gp145 and not the other way around.”

We have changed the subheading to “Recognition of gp145 by broadly neutralizing antibodies 2G12 and PG16”. Thank you for the suggestion.

---

## [Decision Letter · Decision Letter 1]

15 May 2020

PONE-D-20-07724R1

A recombinant gp145 Env glycoprotein from HIV-1 expressed in two different cell lines: effects on glycosylation and antigenicity

PLOS ONE

Dear Dr. Baerga-Ortiz,

Thank you for submitting your manuscript to PLOS ONE. After careful consideration, we feel that it has merit but does not fully meet PLOS ONE’s publication criteria as it currently stands. Therefore, we invite you to submit a revised version of the manuscript that addresses the points raised during the review process.

We would appreciate receiving your revised manuscript by Jun 29 2020 11:59PM. To enhance the reproducibility of your results, we recommend that if applicable you deposit your laboratory protocols in protocols.io, where a protocol can be assigned its own identifier (DOI) such that it can be cited independently in the future. For instructions see: http://journals.plos.org/plosone/s/submission-guidelines#loc-laboratory-protocols

We look forward to receiving your revised manuscript.

Kind regards,

Juan Pablo Jaworski, D.V.M., M.Sc., Ph.D.

Academic Editor

PLOS ONE

Additional Editor Comments (if provided):

Dear authors

Revise your manuscript highlighting strengths and limitations of your current study as suggested by Reviewer 2.

Best,

Dr Jaworski

Reviewers' comments:

Reviewer's Responses to Questions

**Comments to the Author**

1. If the authors have adequately addressed your comments raised in a previous round of review and you feel that this manuscript is now acceptable for publication, you may indicate that here to bypass the “Comments to the Author” section, enter your conflict of interest statement in the “Confidential to Editor” section, and submit your "Accept" recommendation.

Reviewer #1: All comments have been addressed

Reviewer #2: All comments have been addressed

2. Is the manuscript technically sound, and do the data support the conclusions?

Reviewer #1: Yes

Reviewer #2: Yes

3. Has the statistical analysis been performed appropriately and rigorously? 

Reviewer #1: Yes

Reviewer #2: Yes

4. Have the authors made all data underlying the findings in their manuscript fully available?

Reviewer #1: Yes

Reviewer #2: Yes

5. Is the manuscript presented in an intelligible fashion and written in standard English?

Reviewer #1: Yes

Reviewer #2: Yes

6. Review Comments to the Author

Reviewer #1: The authors have addressed all minor and major comments thoroughly. This reviewer thinks that manuscript should be accepted with no further modifications.

Reviewer #2: The author’s explanation of why they used an uncleaved clade C gp145 is satisfactory. These reasons should be stated clearly in the paper to provide proper context to anchor a reasons as to why the work was done - that the product is in a clinical trial so it is useful to know this information. As it stood, there was no explanation for the choice of strain and format. I think the point that it is in clinical trial should even be stated in the abstract, as I really think that will increase reader’s interest which might otherwise be lost due to the uncleaved platform. I don’t believe that uncleaved gp145 is a useful format, regardless of the clinical trial. At least from the perspective of inducing neutralizing antibodies, uncleaved gp140, like gp120 monomer have by now been well-established in dozens of studies (antigenicity and immunogenicity) as poor choices. It may be in both of the clinical trials mentioned in the response letter that neutralizing antibodies are not the end goal and that cellular or non-neutralizing antibodies are instead the goals. If this is the case, however, evaluating the constructs with PG9 and 2G12 is not crucial and they could be assessed with other antibodies that don’t neutralize, like V3.

The statement that gp120 doesn’t work line 61 and “longer” constructs are needed including MPER is a bit vague. This is not about counting epitopes and including them, but more about conformation. It makes more sense to state that forms that better resemble viral spikes are preferable. Uncleaved gp140 is not really much of an improvement on gp120 as it consists of loose gp120 protomers held together by hydrophobic gp41, as shown by EM.

Overall, I would think that some acknowledgement that gp145 is not a cutting-edge platform for inducing neutralizing antibodies should be added somewhere, as otherwise the paper ignores much of the advances in vaccines in preclinical work in recent years. Uncleaved gp145 may have been a reasonable choice in say 2005, but not really in 2020.

All this said, given that this product is in trial justifies the work, but the link to a trial is needed to be stated prominently to justify it.

Regarding the point in the response letter that uncleaved trimers contain more mannose than SOSIP or membrane versions (PMID 26311893 and 26018173), this statement is at odds with the work of Pritchard mentioned directly above this in the letter where there are more complex glycans on uncleaved gp140 (PMID 26051934). This may be just a typo that they meant complex not high mannose glycans. I think this is correct in the main text.

It is good that some previous papers on studying glycans on uncleaved gp140 have been cited in the discussion, but these also need to be covered in the introduction. Glycopeptide mass spec has been done by Crispin, Desaire and others on uncleaved gp140s and I think that is a key part of the introduction missing. Acknowledging this is OK, because the work here is still justified by comparing the uncleaved trimer production from two cell lines, one of which is FDA approved (CHO) and that the gp140 is in clinical trial. So, I am happy that these references were added, but they need to be said earlier to provide context.

Line 94: the uncleaved gp140 is partly trimer, but also a mix of dimer and other not purified, so I think you need to drop the trimer on line 94.

7. PLOS authors have the option to publish the peer review history of their article (what does this mean?). If published, this will include your full peer review and any attached files.

Reviewer #1: No

Reviewer #2: No

---

## [Author Response · Author response to Decision Letter 1]

29 May 2020

Response to Reviewer #1: Thank you for your input, which made this manuscript better and more polished. 

Response to Reviewer #2: We greatly appreciate your insightful comments on the suitability of an uncleaved gp145 as a protective immunogen. Your feedback has given us the chance to reflect on our efforts to produce viral glycoproteins and to discuss possible avenues for the production of protein-based vaccines against SARS-CoV-2.

“The author’s explanation of why they used an uncleaved clade C gp145 is satisfactory. These reasons should be stated clearly in the paper to provide proper context to anchor a reasons as to why the work was done - that the product is in a clinical trial so it is useful to know this information. As it stood, there was no explanation for the choice of strain and format. I think the point that it is in clinical trial should even be stated in the abstract, as I really think that will increase reader’s interest which might otherwise be lost due to the uncleaved platform.” 

We have included a sentence in the abstract and in the introduction to mention the on-going clinical trial, as suggested by the reviewer. And we do agree with the reviewer that the fact CO6980v0c22 is being tested clinically, makes it more relevant to the average reader. However, the objective of our manuscript is to compare biochemical properties for this immunogen expressed in CHO and 293 cells. We are not making any claims of efficacy or immunogenicity for this specific construct. The observations reported by us would still be valid if the gp145 immunogen did not elicit the desired neutralizing humoral response in humans, and could still be useful to researchers expressing other viral glycoproteins for purposes other than vaccine development.

“I don’t believe that uncleaved gp145 is a useful format, regardless of the clinical trial. At least from the perspective of inducing neutralizing antibodies, uncleaved gp140, like gp120 monomer have by now been well-established in dozens of studies (antigenicity and immunogenicity) as poor choices. It may be in both of the clinical trials mentioned in the response letter that neutralizing antibodies are not the end goal and that cellular or non-neutralizing antibodies are instead the goals. If this is the case, however, evaluating the constructs with PG9 and 2G12 is not crucial and they could be assessed with other antibodies that don’t neutralize, like V3.”

The description of the clinical trial (https://clinicaltrials.gov/ct2/show/NCT03382418) clearly states that among the secondary outcomes that will be measured are the response rates and levels of CD8+ and CD4+ T cells, and the response rates and levels of neutralizing antibodies. While the results of these measurements are not presently available, we have chosen not to speculate on what these results might eventually be. We do understand that the reviewer feels strongly that this vaccine is unlikely to yield a neutralizing response. Our manuscript, however, does not make any claims of efficacy or protection, as we focus on the differences in the chemical properties of the immunogen which may result from the choice of expression host.

“The statement that gp120 doesn’t work line 61 and “longer” constructs are needed including MPER is a bit vague. This is not about counting epitopes and including them, but more about conformation. It makes more sense to state that forms that better resemble viral spikes are preferable. Uncleaved gp140 is not really much of an improvement on gp120 as it consists of loose gp120 protomers held together by hydrophobic gp41, as shown by EM.”

Line 61: We do not disagree with this observation and have deleted the phrase: “to preserve known bNAb epitopes within the membrane proximal external region (MPER)” and inserted “that better resemble viral spikes”.

“Overall, I would think that some acknowledgement that gp145 is not a cutting-edge platform for inducing neutralizing antibodies should be added somewhere, as otherwise the paper ignores much of the advances in vaccines in preclinical work in recent years. Uncleaved gp145 may have been a reasonable choice in say 2005, but not really in 2020. All this said, given that this product is in trial justifies the work, but the link to a trial is needed to be stated prominently to justify it.”

We have included a sentence in the abstract and another one in the introduction to mention the on-going clinical trial, as suggested by the reviewer. 

“Regarding the point in the response letter that uncleaved trimers contain more mannose than SOSIP or membrane versions (PMID 26311893 and 26018173), this statement is at odds with the work of Pritchard mentioned directly above this in the letter where there are more complex glycans on uncleaved gp140 (PMID 26051934). This may be just a typo that they meant complex not high mannose glycans. I think this is correct in the main text.”

Yes. The reviewer is correct. There was a mistake in the previous response letter. However, as noted by the reviewer, the manuscript contains the correctly referenced finding that SOSIP or membrane versions of Env, contain a higher proportion of mannose glycans than the uncleaved gp140.

“It is good that some previous papers on studying glycans on uncleaved gp140 have been cited in the discussion, but these also need to be covered in the introduction. Glycopeptide mass spec has been done by Crispin, Desaire and others on uncleaved gp140s and I think that is a key part of the introduction missing. Acknowledging this is OK, because the work here is still justified by comparing the uncleaved trimer production from two cell lines, one of which is FDA approved (CHO) and that the gp140 is in clinical trial. So, I am happy that these references were added, but they need to be said earlier to provide context.”

We have included in the introduction (line 93) the following sentence: “From these detailed glycosylation studies it was concluded that the gp120 portion of the SOSIP trimer contains more high-mannose glycans than the gp41 portion, and that those trimers made in CHO-K1 cells contain a higher level of complex glycosylation than the cleaved trimers produced in 293 cells [45].

“Line 94: the uncleaved gp140 is partly trimer, but also a mix of dimer and other not purified, so I think you need to drop the trimer on line 94.” 

The word “trimer” has been substituted for “Env constructs”.

---

## [Editor Report · Decision Letter 2]

5 Jun 2020

A recombinant gp145 Env glycoprotein from HIV-1 expressed in two different cell lines: effects on glycosylation and antigenicity

PONE-D-20-07724R2

Dear Dr. Baerga-Ortiz,

We’re pleased to inform you that your manuscript has been judged scientifically suitable for publication and will be formally accepted for publication once it meets all outstanding technical requirements.

Kind regards,

Juan Pablo Jaworski, D.V.M., M.Sc., Ph.D.

Academic Editor

PLOS ONE
---

## [Editor Report · Acceptance letter]

10 Jun 2020

PONE-D-20-07724R2 

A recombinant gp145 Env glycoprotein from HIV-1 expressed in two different cell lines: effects on glycosylation and antigenicity 

Dear Dr. Baerga-Ortiz:

I'm pleased to inform you that your manuscript has been deemed suitable for publication in PLOS ONE. Congratulations! Your manuscript is now with our production department. 

Kind regards, 

on behalf of

Dr. Juan Pablo Jaworski 

Academic Editor

PLOS ONE